

**Modeling impacts of climate change and grazing effects on plant**
**biomass and soil organic carbon in the Qinghai–Tibetan**
**grasslands**
Wenjuan Zhang[1,2], Feng Zhang[3], Jiaguo Qi[4], Fujiang Hou*[1]
1 College of Pastoral Agriculture Science and Technology, State Key Laboratory of Grassland Agro-
ecosystems, Lanzhou University, Lanzhou, 730020, China
2 Grassland Management Administration of Qinghai Province, Xining, Qinghai, 810008, China
3 Institute of Arid Agroecology, School of Life Sciences, State Key Laboratory of Grassland Agro-
ecosystems, Lanzhou University, Lanzhou, Gansu, 730000, China
4 Center for Global Change and Earth Observations, Michigan State University, East Lansing, MI
48823, USA
**Abstract:**
The Qinghai Province supports over 40% of the human population but occupies about 29% of
the land area, and thus it plays an important role in the entire Qinghai–Tibetan Plateau (QTP).
The dominant land cover is grassland, which has been severely degraded over the last decade
due to a combination of increased human activities and climate change. Numerous studies
indicate that the plateau is sensitive to recent global climate change, but the drivers and
consequences of grassland ecosystem change are controversial, especially the effects of climate
change and grazing patterns on the grassland biomass and soil organic carbon (SOC) storage in
this region. In this study, we used the DeNitrification-DeComposition (DNDC) model and two
climate change scenarios (representative concentration pathways: RCP4.5 and RCP8.5) to
understand how the grassland biomass and SOC pools might respond to different grazing
intensities under future climate change scenarios. More than 1400 grassland biomass sampling
points and 46 SOC points were collected, which were then used to validate the simulated results.
The results showed that compared with the past 30 years, the biomass and SOC exhibited a
significant decreasing trend under all grazing intensities in the RCP4.5 and RCP8.5 scenarios,
and RCP8.5 had a more negative future effect on the biomass compared with RCP4.5. Thus,
future climate change could lead to greater temporal and spatial variations in the grassland





biomass and SOC. Overall, climate change may be the major factor that leads to fluctuations in
the inter-annual grassland biomass on the Qinghai Province, where the grazing intensity has
significantly affected the grassland vegetation dynamics. Therefore, urgent ecological
conservation of vulnerable grassland ecosystems is required to effectively regulate grazing
practices.
**Keywords:** Biogeochemical process; DNDC; Grazing intensity; Grassland management;
Degradation;

# 37 1 Introduction

Grassland is one of the most widespread terrestrial ecosystems and accounts for nearly 33% of
the land without ice cover (Ellis and Ramankutty, 2008), where it plays important roles in both
the global carbon cycle and terrestrial ecosystem processes (Li et al., 2013c). The Qinghai-
Tibetan Plateau (QTP) about 130 million hectares (ha), it counted 44% of China's grassland
area (Li et al., 2013a; Piao et al., 2012). This area plays a vital role for the ecological services
of China and Southeast Asian countries (Harris, 2010; Li et al., 2013b; Piao et al., 2012; Wang
et al., 2002; Zeng et al., 2015). Qinghai Province supports over 40% of the population but it has
about 29% of the total area, and thus it plays an important role in the whole QTP (Li et al.,
2013a; Piao et al., 2012). This area is recognized as one of the most ecologically fragile and
sensitive areas to global climate change and human disturbance (Harris, 2010; Li et al., 2013b;
Piao et al., 2012; Wang et al., 2002; Zeng et al., 2015). Moreover, this area is also the largest
animal husbandry production region in China, and it also contains the headwaters of the two
major rivers in China, i.e., the Yellow River and the Yangtze River, and thus it plays a vital role
in ecological conservation in China (Zeng et al., 2015).
In recent decades, due to climate change, increased human disturbances, the high altitude alpine
grassland ecosystems, which are the dominant grassland vegetation type, have been severely
degraded (Gao et al., 2010). The air temperature on the plateau has increased by 0.3°C per
decade, which is three times the global average (Li et al., 2008). Warming could significantly
increase the net primary productivity of alpine meadows (Chen et al., 2013; Du et al., 2004;
Fan et al., 2010). Other studies have found that warming also speeds up the decomposition rate

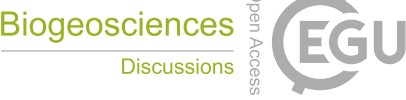



for litter and manure, and increases soil respiration (Luo et al., 2010; Xu et al., 2010), which
could cause significant losses of soil organic carbon (SOC) and affect the alpine grassland
ecosystem carbon pool balance (Pei et al., 2009; Tan et al., 2010). Although the ecological
impact of warming on the QTP alpine grassland ecosystem has not been fully elucidated in
previous studies, there is no doubt that warming will greatly accelerate the key processes in the
alpine grassland ecosystem carbon cycle (Luo et al., 2010). There are reported that both
precipitation amount and the number of precipitation days have increased significantly in QTP
(Li et al., 2010). As precipitation is another crucial climate factor in controlling the carbon cycle
of grassland ecosystems, how the higher variability precipitation impacts the SOC and biomass
in QTP need further investigation (Lehnert et al., 2016; Maussion et al., 2014).
Grazing is the most important biotic factor among the ecological processes that affect rapid
changes in the vegetation and soil, and it is the main method for deriving ecosystem services
from the QTP grassland (Tanentzap and Coomes, 2012). Moreover, grazing is one of the major
human disturbances to the grassland in this area. In general, overgrazing is considered to be one
of the main causes of carbon and nitrogen losses from the soil, thereby contributing to the
unsustainable use of grassland (McIntire and Hik, 2005). Therefore, sustaining a reasonable
grazing intensity has an indispensable role in maintaining the turnover of soil nutrients and
plant community stability (Klein et al., 2007).
Previous studies have shown that different types of vegetation and soil nutrient pools exhibit
significantly different responses to variations in the grazing intensity (Lavado et al., 1996).
However, there is still a lack of robust studies to evaluate the combined effect of grazing and
climate change, as well as their impact on the QTP grassland ecosystem at a large scale. Due to
the unique geographic characteristics and important ecological functions of the QTP grassland
ecosystem, it is necessary to evaluate the impacts of human management and climate change to
ensure that it continues to provide these ecosystem services.
In this study, using a well-calibrated DeNitrification-DeComposition (DNDC) model based on
long-term vegetation observations, we evaluated the response of the grassland ecosystem in
Qinghai Province in terms of both climate change and human management by analyzing the
grazing intensity. We also analyzed the interactions between grassland vegetation and soil
carbon storage with grazing intensity and climate change disturbances at a large scale in long-



term impact assessments.

# 2  Materials and methods

## 2.1  Study area

Qinghai Province (89°35′–103°04′ E, 31°39′–39°19′ N) is located in the northeast of QTP in
China (Fig. 1). This region has a typical plateau climate, with a mean annual temperature of
8.6°C (from –6°C to 9°C) and a mean annual precipitation of 424.7 mm (16.7–776.1 mm). In
general, the climate is cold and dry. The altitude of Qinghai province ranges between 1,650–
6,860 meters above sea level (m a.s.l.) and 67% of the land area is in the range of 3,000–5,000
masl. Grassland is the major land cover in the study area where alpine meadow and alpine
steppe are the dominant vegetation types, where they account for 60.5% of the total grassland
area.
Grazing is the primary human activity in the study area and livestock production is a key
industry in this region. Generally, natural grassland is the major food source for the livestock
in the QTP. Compared with 1949, the number of livestock has increased by almost three times
from $7.49 \times 10^6$ (Zhang, 2011) to the peak number $22.19 \times 10^6$ head in 2005 at the study area
(QPBS, 2015, 2005).
Since 2004, the Chinese government has implemented a series of ecological protection projects
and policies in Qinghai province, including reducing livestock and prohibiting grazing, building
fences to allow natural grassland recovery, as well as providing allowances and awards to local
herdsmen families to promote degraded pasture recovery and to balance the livestock rate
according to the forage productivity (Zeng et al., 2015). The core objective of these projects
and policies is changing the grazing intensity and achieving a balance between the livestock
intensity and grassland regenerability in order to construct a sustainable grassland ecosystem.
Due to new policies for ecological protection, the livestock numbers have declined in recent
years, but they have been maintained at the 2015 level of $19.42 \times 10^6$ head (supplementary
Table S1) (QPBS, 2015).



## 2.2 DNDC model


The DNDC 9.5 biogeochemical model, which was downloaded from the official web
(http://www.dndc.sr.unh.edu/), was employed in this study (Li et al., 2006; Li et al., 1992). The
model has been used widely in more than 20 countries to obtain accurate calibration and
verification results in various ecosystems (Abdalla et al., 2009; Chen et al., 2015;
Kariyapperuma et al., 2011; Li et al., 1996; Li et al., 2017; Li et al., 2014; Liu et al., 2006; Xu
et al., 2003; Zhang and Niu, 2016; Zhao et al., 2016).
The model has two major components. The first component can simulate the soil environmental
conditions, where it includes soil climate, vegetation growth, and decomposition submodels.
The second component includes three submodels for simulating nitrification, denitrification,
and fermentation processes, which are used to simulate biogeochemical production,
consumption, and emissions of $CH_4$, $N_2O$, NO, and $NH_3$, net ecosystem exchanges of $CO_2$, as
well as carbon and nitrogen losses due to leaching (Zhang et al., 2015).
The DNDC model simulates vegetation growth by tracking photosynthesis, respiration, water
demand, N demand, C allocation, crop yield, and litter production. The model predicts the SOC
dynamics mainly by quantifying the SOC input from crop litter incorporation and manure
amendment, as well as the SOC output through decomposition. More detailed information
about the model was given by Li (1996).

## 2.3 Regional database


In order to characterize the spatial heterogeneity of natural grasslands in the study area, we
collected the following geospatial data as inputs for the DNDC biogeochemical model:
grassland type and spatial distribution (Fig. 1), soil properties, and climate data.
**Grassland Database**
The vegetation parameters in the model were obtained from a grassland field monitoring project
implemented during 2005–2014 (ERSMC-a, 2016; ERSMC-b, 2016). This annual monitoring
project covered the major types of grassland within the project area. On average, 168
monitoring sites were sampled each year. For each monitoring site, the average value based on



six replicate sampling points was calculated to determine the aboveground biomass value for
the monitoring site. The aboveground biomass harvests used the quadrat method during the
plant growing season (July 10–August 20) in a 1 m × 1 m plot. A more detailed description of
the sampling method used to obtain the observation data can be found in reports by the
Ecological Environment Remote Sensing Monitoring Center of Qinghai Province (ERSMC-a,
2016; ERSMC-b, 2016). The grassland simulation based on the grassland functional group type
was categorized according to the grassland type map for the study area (Fig. 1). The detailed
grassland parameters used in the model were shown in Supplementary Table S2.

**Soil Database**


We used a 1:1,000,000 scale soil database developed by the Institute of Soil Science, Chinese
Academy of Sciences, which was compiled based on the second national soil survey conducted
in 1979–1994 for all the counties in China (Shi et al., 2004). The database had three attributes:
locations, soil attributes, and reference systems. It contained multi-layer soil properties (e.g.
organic matter, pH, and bulk density), soil texture (e.g. sand, silt and clay proportions), and
spatial information (Shi et al., 2004; Yu et al., 2007a; Yu et al., 2007b), which were used in the
model simulations.

**Climate Database**


Daily climate data were obtained from the China Meteorological Network for the study period,
and there were 39 stations inside the study areas (http://data.cma.cn/). The daily precipitation
and maximum/minimum temperatures between 1985–2014 were interpolated at 1-km
resolution grid for our model. Regression kriging and the inverse distance method were
employed for air temperature and precipitation interpolation, respectively (Fortin and Dale,
2005; Hengl et al., 2007).

**Model implementation**


All datasets were processed with ArcGIS version 10.2 (ESRI, Redlands, CA) to the formation
a georeferenced DNDC regional simulation database. The data processing flowchart could be
found in the supplementary Fig. S1. The county boundary data intersected with grassland type
map to formation the model simulation unit, meanwhile, the county based grazing intensity, soil



properties and climate information also assigned to the model simulation units. The DNDC was
running with regional simulation database based on individual model simulation units. The
detailed information of how to run the model could be found in Li (2012). The actual climate,
soil, grassland type and grazing intensity as the simulation baseline.

## 173   2.4   Simulation scenarios

**Grazing simulation scenarios**
The grazing period is all-year round and cattle (90% yaks), sheep, and goats are major livestock
types, while horses are a minor component in the study area. The grazing intensity data were
based on the annual national livestock statistical report provided by the National Bureau of
Statistics of China and the Bureau of Statistics for Qinghai Province. The detailed grazing data
are shown in Supplementary Table S3. In the DNDC model, grazing activity is defined by
specifying the grazing parameters, including the livestock type, grazing period, and grazing
intensity. The detailed parameters for the DNDC grazing model are shown in Supplementary
Table S4. The grazing intensity is defined according to Eq. 1 based on the grazing area in each
administrative region (Li et al., 2014):

$$GI = LP/GA, \qquad\qquad (Eq.1)$$

where GI is the grazing intensity (head ha$^{-1}$), LP is the livestock unit (head), and GA is the
grazing area (ha).
In order to test the responses of the grassland biomass and soil SOC to various grazing
intensities, we tested the following treatments: baseline, grazing intensity based on the actual
grazing intensity in 2005; $G_0$, grazing intensity of zero; $G_{-50}$, 50% of the baseline intensity; and
$G_{+50}$, 50% higher than the baseline.
**Climate change scenarios**
The Intergovernmental Panel on Climate Change (IPCC) Fifth Report employed new stable
concentration-based scenarios in representative concentration pathways (RCPs) to project
future climate change (IPCC, 2013). The development of the RCP scenarios used a parallel
method, which combined climate, air, and the carbon cycle with emissions and the socio-
economic situation to assess the impact of climate change on a study area, as well as adaptation,

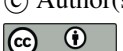



vulnerability, and mitigation analysis (Moss et al., 2010). The RCPs were named according to
their 2100 radiative forcing level and reported by individual modeling teams, i.e., 2.6–8.5 W/m$^2$.
The RCPs comprise four scenarios, i.e., RCP2.6, RCP4.5, RCP6.0, and RCP8.5 (Moss et al.,
2010). Each scenario provides a path affected by social and economic conditions and climate,
and each projection corresponds to the radiation force value predicted by 2100.
We considered RCP4.5 and RCP8.5 because these two scenarios have been used widely to
evaluate the potential impact of climate change on the environment (Di Vittorio et al., 2014; Li
et al., 2015; van Vuuren et al., 2011; Zhang et al., 2013). RCP4.5 represents a medium-low RCP
with stabilization of $CO_2$ emissions from 2150 onwards, and RCP8.5 represents a high RCP
with stabilizing $CO_2$ emissions post-2100 (Meinshausen et al., 2011). The projected climate
conditions in the present study under RCP4.5 and RCP8.5 were derived from the average values
of 25 Coupled Model Intercomparison Project Phase 5 (CMIP5) global climate models (Fu and
Feng, 2014).
Compared with 2014, the average temperature and precipitation increased by 0.72°C and
0.80°C, and by 11.81 mm and 12.50 mm under RCP4.5 and RCP8.5 in 2044, respectively, in
the study area (Table 1). The changes in the spatial distribution of precipitation are shown in
Supplementary Fig. S2. The pattern of increased precipitation was similar using RCP4.5 and
RCP8.5 for the period of 2014–2044, where it increased in the whole area and it increased
gradually from the north to the south of the study area. However, RCP8.5 obtained a higher
increase than RCP4.5 and the southwest part of the research area is projected to have a higher
temperature increase than the other regions. Moreover, the annual average temperature had a
similar distribution under the two climate change scenarios, where the temperature increase
using RCP4.5 (Supplementary Fig. S2c) was lower than that with RCP8.5 (Supplementary Fig.
S2d).
Three different periods were considered in the grassland simulations. First, a pretreatment
(1961–1984) period was used to initialize the soil climate conditions and SOC composition.
The pretreatment period represented the baseline climate with no increases in $CO_2$ or climate
change. The second period represented realistic climate scenarios (1985–2014) based on the
most recent climate. The third period comprised future climate scenarios (2015–2044), which
represented two future climates (RCP4.5,RCP8.5) scenarios with changes in temperature and



precipitation.

## 2.5    Model validation

The root mean squared error (RMSE) (Eq.2), coefficient of determination ($R^2$) (Eq.3) and model
efficiency (ME) (Eq.4) were employed for model validation. The RMSE estimates the scatter
between the simulated and measured data, where values close to zero indicate excellent
agreement and hence the good performance of the model (Araya et al., 2015). $R^2$ is used to test
the agreement between the modeled results and observations, where a value closer to 1 indicates
that the model provides a better explanation for the observed values (Willmott, 1982). The
positive ME value indicates that the model prediction is better than the mean of observations,
and the best model performance has ME value equal to 1 (Miehle, 2006). RMSE, $R^2$ and ME
were calculated as follows:

$$\text{RMSE} = \sqrt{\frac{\sum_{i=1}^{n}(P_i - O_i)^2}{n}} \qquad \text{(Eq. 2)}$$

$$R^2 = \left[\frac{\sum_{i=1}^{n}(O_i - \bar{O})(P_i - \bar{P})}{\sqrt{\sum_{i=1}^{n}(O_i - \bar{O})^2}\sqrt{\sum_{i=1}^{n}(P_i - \bar{P})^2}}\right]^2, \qquad \text{(Eq. 3)}$$

$$\text{ME} = 1 - \frac{\sum_{i=1}^{n}(P_i - O_i)^2}{\sum_{i=1}^{n}(O_i - \bar{O})^2} \qquad \text{(Eq. 4)}$$

where $P_i$ and $O_i$ were modeled and observed values, and $\bar{P}$ and $\bar{O}$ are their averages. n is the
number of values.
The validation dataset included more than 1400 grassland biomass sampling points, which
covered the whole of the study area, and the field measurements were also fully representative
of the major grassland types in this area. In addition, 46 SOC observation points were sampled
between 2011−2012, which were randomly distributed among all of the simulation units
(county and grassland types). The grassland biomass was sampled in quadrat (1 m × 1m) with
3 replicates between mid-July and mid-August. Maximum biomass in each quadrat was
harvested and dried in an oven at 70 °C for 72 h, weighed and ground for analysis. The soil of
0−30 cm depth was sampled at 10-cm intervals with a soil drill (metal cylinder: diameter of 5
cm, length of 20 cm and the total length of the sampler 1.3 m). 3 samples were collected in each
replication plot. The ground soil samples passed a 0.15-mm sieve and wet oxidation method





was applied to determine SOC (Mebius, 1960). In general, every simulation unit had 1–2
validation points (ERSMC-a, 2016).

## 2.6   Statistical analysis

Two-way analysis of variance (ANOVA) was used to test the effects of climate and grazing
intensity on both the biomass and SOC. Mean values for the same treatments were compared
using Fisher's least significant difference (LSD) test with one-way ANOVA at $P = 0.05$. The
statistical analyses, including the test for normality (Shapiro-Wilk) and homogeneity of
variance (Levene), were performed using Origin 2016 version b9.3.1.273 (OriginLab
Corporation, MA, USA), and the multiple regression analysis was conducted with the Minitab
version 17 (Minitab Inc., State College, PA, USA).

## 3   Results

### 3.1   Model validation

The biomass simulation showed that the modeled total biomass was in good agreement with
the observations (Fig. 2). There was a significant linear relationship ($P < 0.001$) between the
measurements and the modeled total biomass ($R^2 = 0.71$, ME=0.75, RMSE = 93.11 g C m$^{-2}$; P
$< 0.001$). The simulated SOC concentrations were in good agreement with the measured data
(Fig. 3). The calculated statistical indices indicated that the modeled SOC concentrations were
closely correlated with the measured data ($R^2 = 0.73$, ME=0.69, RMSE = 21.51 g C kg$^{-1}$; $P <$

267   0.001).

### 3.2   Sensitivity analysis

A series of sensitivity tests were conducted to investigate the responses of the DNDC to
variation in climate factors (air temperature, precipitation) and grazing intensity. DNDC was
run with a 55-year baseline scenario that was based on the actual climate, soil and grazing
conditions of year 2005 in the study area. The ranges of values for alternative scenarios were




±10, ± 20 and ±30% for precipitation, ±1, ± 2 and ±3 °C for air temperature and ±20, ±40, ±60,
±80 and ±100% for grazing intensity, respectively.
In the sensitivity analysis simulation, increases in precipitation resulted in elevated biomass and
SOC, however, the SOC was changed slightly compared to the biomass (Fig. 4A, B);
Temperature decrease induced the biomass decrease, and temperature increase could increase
the biomass. However, biomass change did not follow a simple linear relationship with change
in temperature. The 1°C temperature increase could bring 24% of biomass increase, meanwhile,
1°C temperature decrease could decrease 13% biomass (Fig. 4A) . Biomass was not susceptible
to the changes in precipitation. The biomass increased 7% and decreased 6% with precipitation
increased and decreased 30%, respectively. SOC had the reverse trend with increased or
decreased temperature, but there was a more complex relationship with temperature change.
The SOC had less sensitivity to temperature change compared to biomass. With a 1 °C
temperature increase, the SOC increased slightly with 0.26%, but when temperature increased
over 2 °C, the SOC decreased 0.26–0.83% (Fig. 4B). The modeled biomass was sensitive to
grazing intensity and biomass had a reverse trend with increased or decreased grazing intensity
(Fig. 4A). When grazing intensity changed from -100 to 100%, SOC increased rate from -0.22
to 0.40% (Fig. 4B). The sensitivity analysis demonstrated that the DNDC model was sensitive
to precipitation and temperature change and was useful for studying the biomass and SOC under
the grazing intensity change.

## 3.3   Impact of grazing on biomass and SOC

The biomass and SOC were significantly affected by climate change and the grazing intensity.
However, there were no significant interaction effects between climate and grazing intensity on
biomass and SOC during 1985–2044 throughout the study area (Table 2).
Under the same climate scenario, the grazing intensity change could significantly influence the
biomass, which had a negative relationship with the grazing intensity. The biomass differed
significantly under the four grazing intensities in the three climate scenarios. Among the
grazing intensity treatments, the biomass followed the order of: G0 > G–50 > baseline > G+50
(Table 3). Compared with the treatment without grazing, the grazing scenarios induced similar

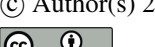



changes in the biomass among the different grazing intensity treatments.
Grazing could increase the SOC storage. The SOC levels under various grazing intensities
followed the order of: $G_0 < G_{-50} < $ baseline $< G_{+50}$ (Table 4). $G_0$ had the lowest SOC whereas
$G_{+50}$ had the highest SOC under all the climate scenarios. Under the same climate scenario, a
reduction in the grazing intensity from the baseline could significantly decrease the SOC
concentration, but there was no significant change in the SOC when the grazing intensity
increased by 50% compared with the baseline.

## 3.4    Impact of climate change on biomass and SOC

The biomass exhibited a significant decreasing trend in the future climate scenarios compared
with the past 30 years under all the grazing intensities (Fig. 5), although precipitation increased
under both RCP4.5 and RCP8.5 (Table 1). Moreover, with the same grazing intensity, the
biomass was lower in RCP8.5 compared with RCP4.5. However, the biomass did not differ
significantly between RCP4.5 and RCP8.5 under the same grazing intensity (Table 3). This
suggests that RCP8.5 had a more negative effect on the biomass compared with RCP4.5 (Fig.

315    5).

The future climate could significantly decrease the SOC, and RCP8.5 had a more negative effect
than the RCP4.5 on the SOC. SOC exhibited a continuously decreasing trend according to the
RCP4.5 and RCP8.5 projections in the research area, where the changes in the SOC were similar
under the different grazing treatments (Fig. 6). A similar trend also occurred between 1985–
2014. The SOC was lower under RCP8.5 compared with that under RCP4.5 at all of the grazing
intensities. However, there were no significant differences between RCP4.5 and RCP8.5 (Table

322    4).

## 3.5   The relationship between SOC and biomass change with grazing and climate factors

A multiple linear regression analysis indicated precipitation, air temperature and combined with
grazing intensity, can explain 33.2% of changes in biomass under the realistic climate scenarios



with a linear model. Meanwhile, precipitation, air temperature, and grazing intensity can
explain 52.3% of SOC variation (Table 5). Taking into account the prediction sum of squares
(PRESS) value, air temperature is the best predictor factor for biomass and SOC. It's suggested
that precipitation and grazing intensity with lower contributes to biomass and SOC change in
study region during past thirty years compared to temperature.

## 4  Discussion

### 4.1   Effects of climate change on biomass and SOC

Climate change is the main driver of the inter-annual fluctuations in the grassland biomass, as
observed in previous studies by Fan et al. (2010) and Gao et al. (2016). The unique climate
conditions such as precipitation and temperature on the QTP have a significant impact on the
grassland biomass (Fan et al., 2010; Yan et al., 2015). According to this study, the biomass of
alpine grassland could increase significantly in the short term as the temperature increases (Fig.
4), as also suggested by Chen et al. (2013) and Gao et al. (2016). However, under long-term
constant warming and without considering other meteorological factors, the alpine grassland
biomass will probably decrease (Zhu et al., 2016). This may be due to the higher temperature
increasing evaporation in the study area, thereby overcoming the benefits of increased
precipitation (Xu et al., 2009). The shortage of water will ultimately limit the increase in the
grassland biomass with significant warming and drying.
The decline of the SOC in our study indicates that climate warming will have more negative
effects and eliminated the positive effect of precipitation increasing in the study area. Riedo et
al. (2000) indicated that carbon storage may be lost from grazed grassland as the temperature
and precipitation increase. Tan et al. (2010) suggested that after a 2°C increase in temperature
in the QTP, the grassland ecosystem's net primary productivity will increase by 9%, but the
SOC will decrease by 10%. Temperature and precipitation are the main factors that affect the
SOC pools (Jobbagy and Jackson, 2000). Many studies have shown that sustained warming will
lead to increases in the SOC decomposition rate (Tan et al., 2010; Xu et al., 2012), especially
in the QTP region with high carbon storage at a low temperature in the high latitudes. Thus, the



SOC could be released by climate warming and become a more obvious carbon source
(Kirschbaum, 1995; Kvenvolden, 1993; Qin et al., 2014; Wang et al., 2008; Yang et al., 2008).
However, the effects of warming and precipitation on SOC storage remain a relatively complex
problem (Cao and Woodward, 1998; Schuur, 2003).

## 4.2   Effects of grazing intensity on biomass and SOC

The grazing intensity is most importance for the outcomes of grazing and it is the main external
factor that controls the influence of the grassland vegetation dynamics, as reported in the
previous studies (Guevara et al., 1996; McIntire and Hik, 2005; Pei et al., 2008; Veen et al.,
2012; Zeng et al., 2015). Indeed, an increase in the grazing intensity implies that more plants
would be removed by animals, which could eventually lead to a decline in the aboveground
biomass of the grassland (Yan et al., 2013).
Small differences in the SOC concentrations were observed after the grazing intensity increased.
However, there was a positive correlation between the grazing intensity with SOC. There is a
lack of consistent conclusions regarding the impact of grazing on the SOC concentration
according to previous studies. Thus, some studies showed that the grazing intensity and SOC
had a negative correlation (Bagchi and Ritchie, 2010; Derner et al., 1997; Wu et al., 2009) or
no relationship (Holt, 1997; Milchunas and Lauenroth, 1993). By contrast, many other studies
showed that grazing can increase the SOC (Li et al., 2011; Schuman et al., 1999; Wienhold et
al., 2001). This is partly because moderate grazing can increase the grassland below-ground
biomass, which is beneficial for the accumulation of SOC (López-Mársico et al., 2015). Some
studies have shown that increasing the plant root/shoot ratio and allocating more carbon to the
root system could induce different increases in the SOC (Derner et al., 1997). Nevertheless, the
main reason for the increase in the SOC in our study was the increasing number of grazing
animals, and thus the increased amount of manure returned after grazing on grassland (Hu et
al., 2015). Furthermore, the fertilizing effects of livestock excrement can increase the SOC
(Conant et al., 2001), especially in alpine grassland where the low temperature leads to the
relatively slow decomposition of litter (Davidson and Janssens, 2006). Moreover, increases in
the effects of hoof activity can accelerate the decomposition of litter and decaying roots, and





improve the contact with the soil, thereby accelerating the transfer of carbon to the soil to
improve the SOC concentration (Luo et al., 2010; Naeth et al., 1991).

## 4.3    Patterns of regional change in the biomass and SOC

From a spatiotemporal distribution perspective, the distribution of grassland biomass in
Qinghai Province is rather distinct due to the different constraints imposed by water and the
cumulative temperature. The biomass increased in the central and southwest of the research
region but decreased in the eastern and northern regions under RCP4.5 and RCP8.5,
respectively. However, the grassland biomass tended to decrease in more regions rather than
exhibiting an increasing trend (Fig. 7A). In particular, the vegetation activities are mainly
controlled by temperature in the eastern region, which may lead to greater negative effects than
the positive effects of increased precipitation (Zhou et al., 2007); therefore, the average regional
biomass may exhibit a significant decreasing trend.
In general, the SOC decreased from the low-temperature region to the high-temperature region,
where it followed the temperature distribution pattern in Qinghai Province and decreased from
the south to the north (Fig. 7B). The cold weather conditions would limit decomposition process
and there would be greater carbon storage over the years with accumulation in this area.
Furthermore, on the regional scale, although the SOC exhibited a decreasing trend in the whole
study area, the rate of change differed with a significant spatial distribution pattern.

## 4.4    Uncertainty analysis

Models are ideal tools for assessing the details of environment processes under various grazing
intensity. Furthermore, they can provide projections regarding the variations in grassland
biomass and SOC under alternative climate change scenarios. However, the uncertainty of the
data sources should be incorporated into the model outputs. The CMIP5 RCP scenarios were
used to provide the possible changes in climate in this study, but as a long-term climate
projection, the uncertainty of the projected climate will increase the time span (Moss et al.,
2010). The precipitation seasonal distribution pattern is critical to grassland growth (Shen et al.,
2011). In the present study, the precipitation distribution pattern of RCP scenarios was derived



from the year of 2014; this assumption may incorporate uncertainty for long term study.
In the present study, we assumed that the grassland type was the same in the scenarios, but the
grassland community structure could be modified by climate change. Therefore, the results of
this study only reflect the impacts of future climate change on the biomass and SOC storage.
The root/shoot ratio is one of the factors that are most sensitive to the simulated results and it
could be a potential source of uncertainty in biomass simulations. In this study, we used the
root/shoot ratio based on the grassland classes, which could generalize the spatial root/shoot
properties in the study area.
The grassland community structure could also be altered under both grazing and climate change
(Koerner and Collins, 2014), so the model assumed that the grassland community structure
remained the same throughout the whole simulation process, and thus it could incorporate
uncertainty in the simulation results. Due to a lack of data regarding the response of grassland
soil to animal trampling in the DNDC model, we ignored the trampling effect of the animals on
the soil structure, which may have led to some errors in the results.
The grazing rate can be another potential source of uncertainty. In most of the natural grassland
regions of the QTP, transhumance is usually practiced, which requires the transfer of livestock
from one pasture to another during different seasons, and staying in the same pasture for the
whole season. However, this grassland management practice was simplified in the present study
because we could not find specific statistical data to address this issue. Thus, we assumed that
livestock stayed in the same pasture for the whole year with 24 h d$^{-1}$ of grazing and the stocking
rates were the same throughout the simulation unit and without yak dung remove (Zhang et al.,
2016). Furthermore, we assumed that all grasslands were useable. These assumptions could
have induced slight uncertainties in the simulation results.

## 5  Conclusions

In this study, we used the DNDC model to study the grassland biomass and SOC dynamics
under different climate change and grazing management scenarios. We found that the biomass
and SOC were significantly affected by climate change and grazing intensity. In the long term,
the total grassland biomass had a negative relationship and the SOC had a positive relationship



with the grazing intensity. The total biomass exhibited interannual fluctuations in the time series
and the SOC had a declining trend. All of the grazing scenarios obtained similar patterns of
change compared with the baseline scenario.
Future climate change could induce great uncertainty in the grassland dynamics. The total
grassland biomass and average SOC in the study area were reduced significantly under both the
RCP4.5 and RCP8.5 future climate change scenarios. However, there were significant
differences in the spatial distribution of the changing trends in the biomass and SOC. In the
eastern and northern regions of the study area, the biomass decreased, whereas it exhibited an
increasing trend in the southwest part of the research area. On a regional scale, the change in
the SOC had a significant spatial distribution pattern where it decreased from the south to the
north.
The grassland biomass and SOC will decline under sustained warming according to future
climate change projections. Therefore, grassland management should be adapted to potential
climate change to ensure sustainable grassland development in the study area. In the future,
suitable grazing intensity for the sustainable development of grasslands should be studied.
Moreover, greater human activity and management practices should be coupled according to
the model to develop more intelligent grassland management strategies.

**Acknowledgments**
We thank editors and four anonymous reviewers for their valuable comments and suggestions
on the manuscript. This study was supported by the National Natural Science Foundation of
China (Nos. 31672472 and 31200335), National Key Project of Scientific and Technical
Supporting Programs (2014CB138706), and Program for Changjiang Scholars and Innovative
Research Team in University (IRT17R50). We are grateful to the grassland station in Qinghai
province for providing data about the grassland biomass and livestock numbers in each county.

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





Table 1. Projected climatic changes (precipitation and maximum, minimum, and mean air temperature) under the RCP4.5 and RCP8.5 scenarios in 2044 compared with the corresponding values in the baseline data (2014).

| Scenarios | Air temperature (°C) | | | Precipitation(mm) |
|---|---|---|---|---|
| | $T_{max}$ | $T_{min}$ | $T_{mean}$ | |
| Baseline | 3.63 | -16.88 | -3.56 | 279.24 |
| RCP4.5 | +0.99 | +0.44 | +0.72 | +11.81 |
| RCP8.5 | +1.09 | +0.51 | +0.80 | +12.50 |

Table 2. Summary of two-way analysis of variance for biomass and SOC relative to the climate, grazing intensity, and their interactions during 1985–2044. Degrees of freedom (d.f.), mean squares (M.S.), variance ratio (F-value), and level of significance (P-value) are shown.

| Source of variation | d.f. | Biomass | | | SOC | | |
|---|---|---|---|---|---|---|---|
| | | M.S. | $F$-value | $P$-value | M.S. | $F$-value | $P$-value |
| Climate | 2 | 16827.91 | 54.27 | ** | 468.16 | 723.54 | ** |
| Grazing Intensity | 3 | 22132.64 | 71.37 | ** | 17.29 | 26.72 | ** |
| Climate*Grazing Intensity | 6 | 2.63 | 0.01 | n.s. | 0.28 | 0.28 | n.s. |

**Significant effect; n.s., no significant effect.

Table 3. Simulated total biomass (mean ± SE) under the realistic, RCP4.5 and RCP8.5 scenarios for the $G_0$, $G_{-50}$, baseline, and $G_{+50}$ treatments.

| Management practice | Total biomass (g C m$^{-2}$) | | |
|---|---|---|---|
| | Realistic (1985–2014) | RCP4.5 (2015–2044) | RCP8.5 (2015–2044) |
| $G_0$ | 223.6±10.8[a A] | 207.0±10.3[a B] | 199.5±10.2[a B] |
| $G_{-50}$ | 213.3±10.5[b A] | 197.3±10.1[b B] | 189.7±10.1[b B] |
| Baseline | 202.1±10.4[c A] | 186.5±10.0[c B] | 178.9±10.0[c B] |
| $G_{+50}$ | 190.2±10.4[d A] | 173.9±10.0[d B] | 166.3±10.0[d B] |

SE: Standard error. Significant differences among management practices are indicated by letters. Values within a column followed by the same lowercase letters or within a row followed by the same uppercase letter are not different at P < 0.05.

Table 4. Simulated SOC (mean ± SE) under the realistic, RCP4.5, and RCP8.5 scenarios for the $G_0$, $G_{-50}$, baseline and $G_{+50}$ treatments.

| Management practice | Soil organic carbon (0–20 cm) concentrations (g C kg$^{-1}$) | | |
|---|---|---|---|
| | Realistic (1985–2014) | RCP4.5 (2015–2044) | RCP8.5 (2015–2044) |
| $G_0$ | 65.84±2.30 [a A] | 62.94±2.20[a B] | 62.87±2.20[a B] |
| $G_{-50}$ | 66.04±2.30 [a A] | 63.27±2.21[a B] | 63.20±2.20[b B] |
| Baseline | 66.32±2.31[b A] | 63.61±2.21[b B] | 63.53±2.21[b B] |
| $G_{+50}$ | 66.59±2.31[b A] | 63.96±2.22[b B] | 63.88±2.21[b B] |

SE: Standard error. Significant differences among management practices are indicated by letters. Values within a column followed by the same lowercase letters or within a row followed by the same uppercase letter are not different at $P < 0.05$.



Table 5. Multiple linear regression analysis of grassland biomass and SOC change with relative factors.

|  | Variables numbers | R-square | PRESS | Temperature | Precipitation | Grazing Intensity |
|---|---|---|---|---|---|---|
| **Biomass** | 1 | 26. 4 | 273067.7 | X | | |
| | 1 | 6.4 | 370402.4 | | | X |
| | 1 | 0.4 | 349337.6 | | X | |
| | 2 | 26.4 | 287817.3 | X | X | |
| | 2 | 26.4 | 301908.4 | X | | X |
| | 2 | 8.6 | 383224.5 | | X | X |
| | 3 | 26.4 | 326183.5 | X | X | X |
| **SOC** | 1 | 47.6 | 179.2 | | X | |
| | 1 | 2.3 | 310.9 | | | X |
| | 1 | 0.4 | 322.9 | X | | |
| | 2 | 47.9 | 185.5 | | X | X |
| | 2 | 47.7 | 189.5 | X | X | |
| | 2 | 4.7 | 328.8 | X | | X |
| | 3 | 48.6 | 199.1 | X | X | X |

PRESS: The prediction sum of squares. The smaller the PRESS value, the better the model´s predictive ability.

X: Indicates variable applied in the regression.





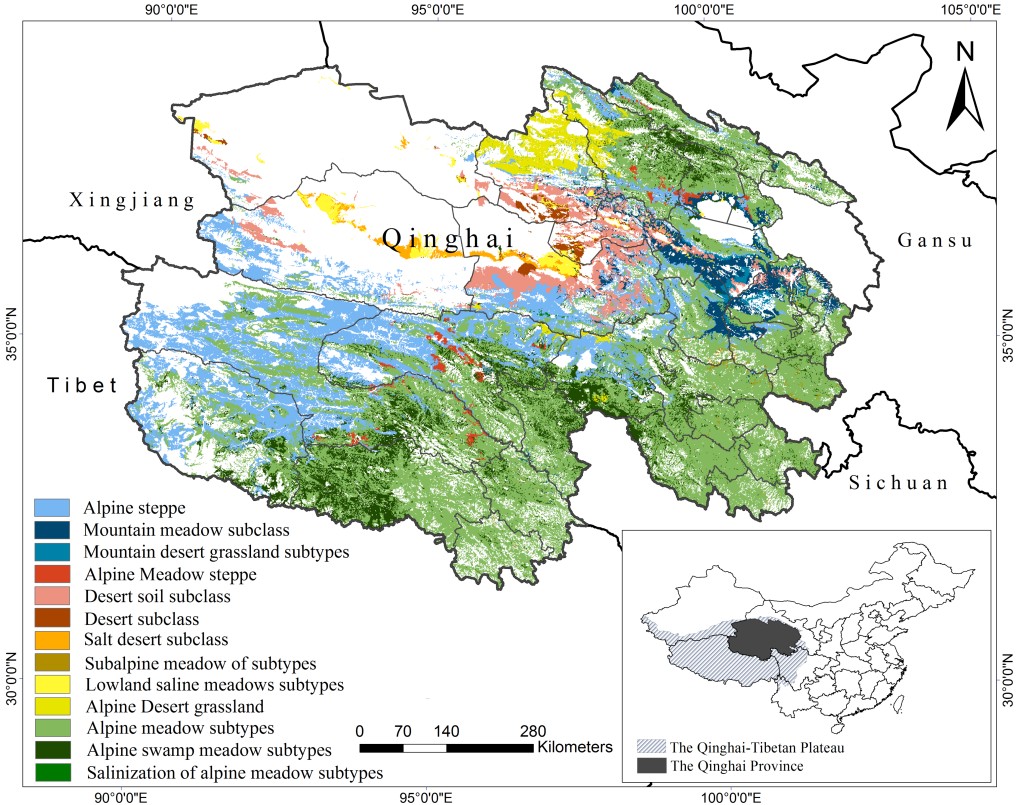

Fig. 1. Location of the study area and spatial distribution of the main grassland types. White areas are not covered by grassland.



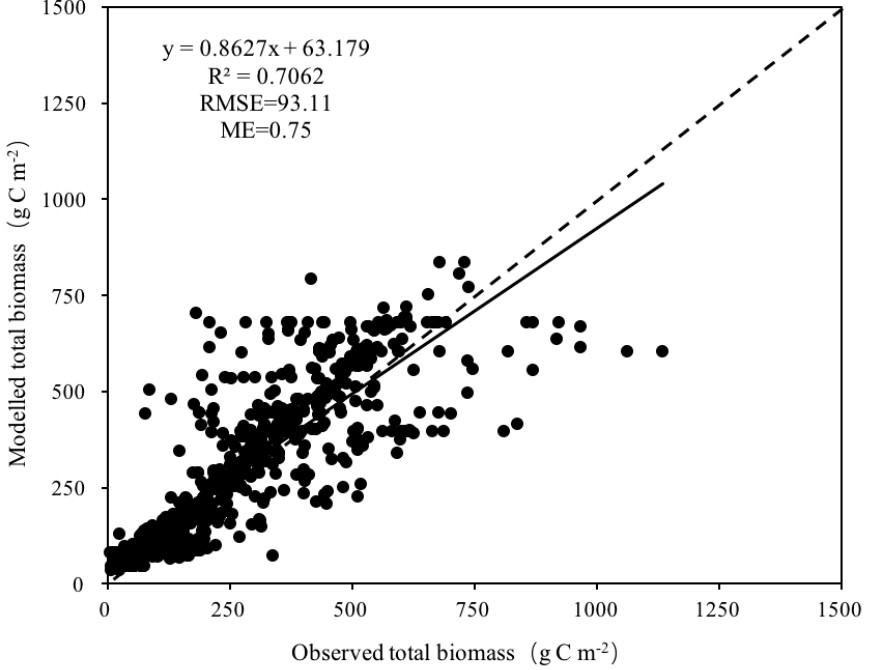

Fig. 2. Comparison of the modeled and observed total biomass values.

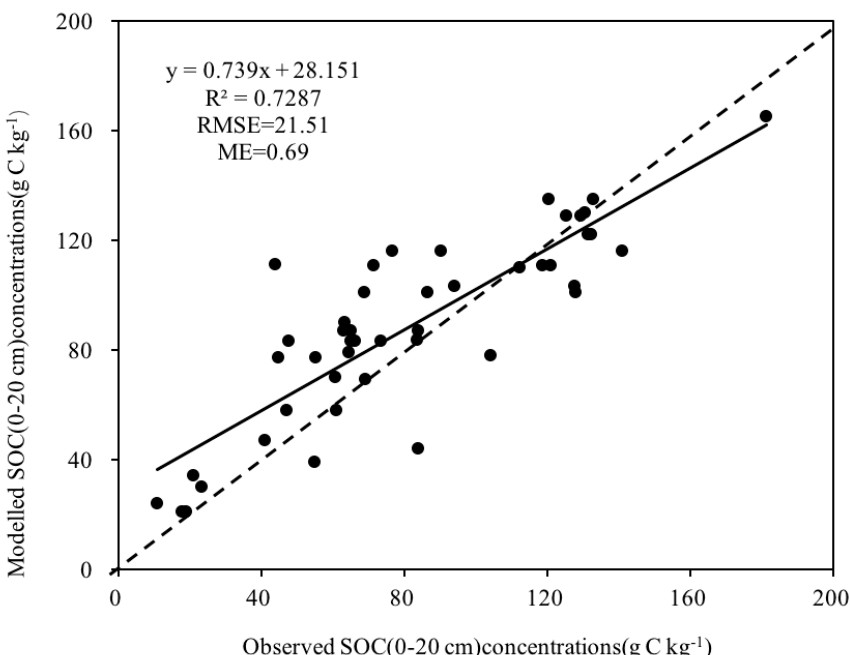

Fig. 3. Comparison of the modeled and observed SOC concentrations (0–20 cm).







Fig. 4. Sensitivity analysis of model response to climate and grazing intensity change. The baseline biomass and SOC were the average value of a 55-year (1961-2014) simulation based on the actual climate and grazing conditions in the study area.




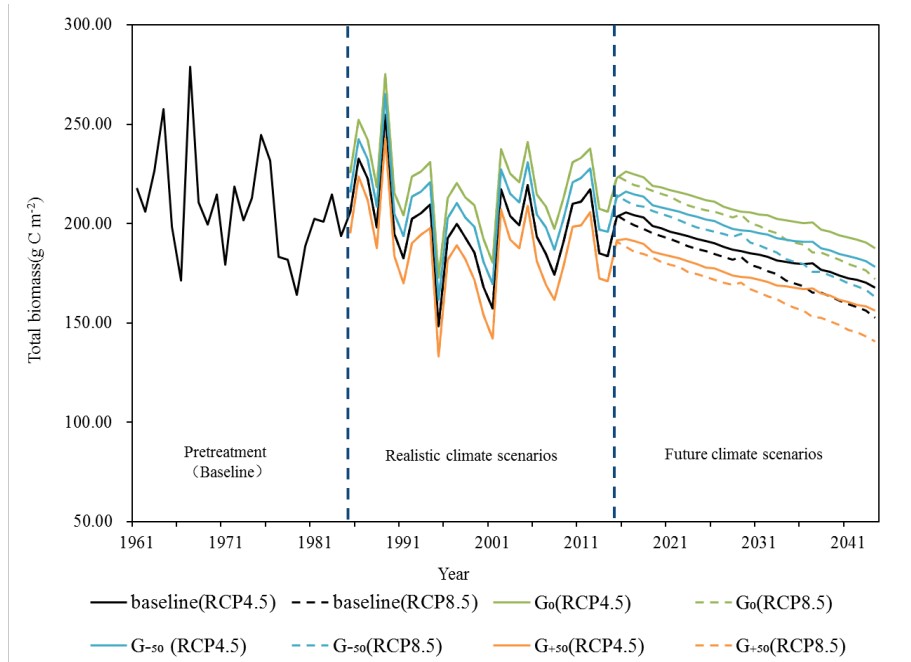

Fig. 5. Variations in the area-weighted mean biomass value under different scenarios. The stage on the left represents the preprocessing period from 1961 to 1984. The stage in the middle represents the realistic climate scenarios. The stage on the right represents future climate scenarios.



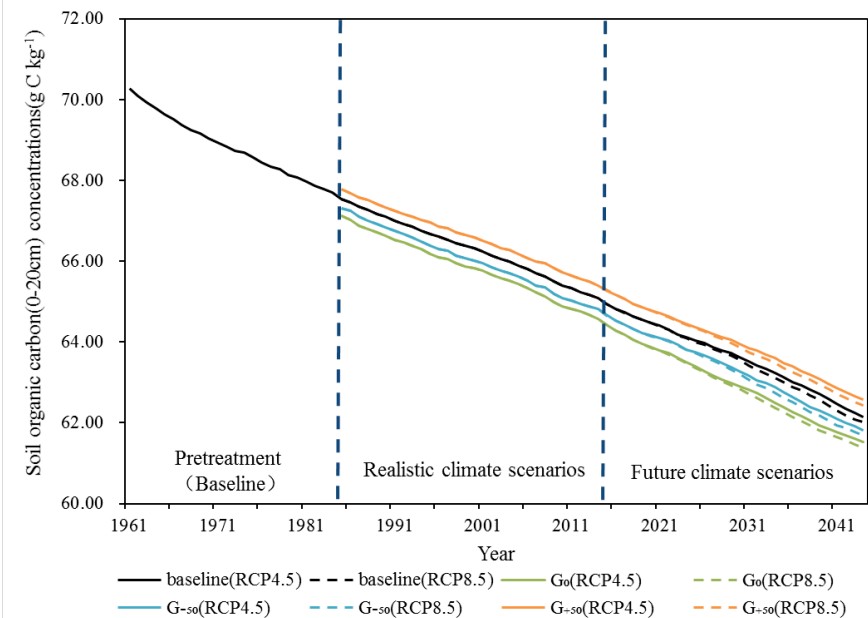

Fig. 6. Variations in the area-weighted mean SOC value under different scenarios. The stage on the left represents the preprocessing period from 1961 to 1984. The stage in the middle represents the realistic climate scenarios. The stage on the right represents future climate scenarios.



Fig. 7. Responses of the grassland biomass(A) and SOC(B) to climate change at a regional scale.