# Peer review of "Modeling impacts of climate change and grazing effects on plant"

_Biogeosciences, 2017_

## Referee Comment (RC1) · Anonymous Referee #2 · 23 Aug 2017

General comments: This study evaluated responses of grassland ecosystems to both climate change and changes of grazing intensity using a biogeochemical model, DNDC. This topic is important, especially for the region of Qinghai–Tibetan Plateau (QTP), considering that vegetation is sensitive to changes in climate/human management and soil organic carbon (SOC) content is high and potentially easy to loss in this area. This study conducted a regional simulation by coupling the DNDC with a database, and the results provided some useful information regarding future changes of biomass and SOC in QTP. However, I have several concerns about this study. The

first concern is about the model application and the results reported (such as the results in Tables 2, 3 and 4; see the specific points). Detailed introduction regarding how the simulations were implemented and what are the reported values in tables are necessary because these contents are necessary for a correct interpretation. The second concern is the statistical analysis in this study. The authors did some statistical analysis to evaluate the simulations. My general feeling is that the statistical analysis could lead to an over-interpretation for the simulated results (if my understanding is correct). In this study, variations of simulations are totally resulted from changes in input parameters (such as climate condition and grazing density), instead of any random factors or instrument errors. Because the authors changed climate condition/grazing density that resulted in the variations, it is not surprising that the biomass and SOC were affected by climate and grazing intensity. So, the descriptions of 'significant effect' are somewhat over-interpretations for me. I suggest the authors rethink about the statistical analysis. The authors may need to clarify that the statistical analysis is not like the statistical analysis for observations with random factors/errors to avoid over-interpretation. Finally, I noticed some inaccurate descriptions, mistakes, and grammar errors.

Specific points: Line 28: Delete 'future'. And I suggest delete 'Thus', because it looks like there is no any causal relationship in these two sentences. Line 41: Grammar error in this sentence. Line 93: Are these ranges spatial variations or temporal variations of multi-years? Line 121: Delete 'major'. Lines 123-126: This is not an accurate description. For example, NEE is primarily simulated by tracking vegetation growth and SOC decomposition (instead of nitrification, denitrification or fermentation) in DNDC. Line 148: Here 'Table s2' should be 'Table s4'. Lines 167-169: Grammar errors in this sentence. Line 179: Are these parameters in the Table s3 DNDC default values, or you determined these parameters based on local information? Line 181: '... for the DNDC grazing model...' should be '... for simulating grass growth'. Lines 224-226: Here, could you please specify how did you build the climate data from 2015 to 2044? I notice that there don't have biomass fluctuations between 2015 and 2044, so I guess there is no

rainfall fluctuation between 2015 and 2044 (i.e. no dry vs. wet years). Did you use 2014 (or another year) as a 'base', then add temperature and precipitation changes to build climate data from 2015 to 2044. The model's behaviors may be largely regulated by initial or base conditions, so a detailed description on how the climate data were built is necessary. Line 244: '3 replicates' here but '6 replicates' at the Line 141. In addition, it is repeated description of the Lines 140 to 143. Line 263 and Fig 2: Total biomass or above ground biomass? In the 'grassland database', you mentioned that 'above ground biomass' is available for model validation. Lines 269-274: This part should be in the 'M&M' section for me. Lines 289-291: This sentence is general and not informative for summarizing the results of the sensitivity analysis. I suggest delete this sentence. Line 295: Could you please specify the variances in Table 2? Did the variances include both the inter-annual variations during 1985 to 2044 and the variations due to grazing intensity change? Lines 300 and 303: The explanations for Tables 3 and 4 are poor. Please clearly explain what are the values in these tables as this influences a correct interpretation. For example, are the values spatial-temporal means across the regions and years (such as 1985-2014 for 'realistic'), with SE representing spatial variations (or other variations)? If so, it may not be fair comparisons between realistic and RCP scenarios because they have different initial conditions (i.e. 'realistic' has a soil condition in 1985 while 'RCPs' have a soil condition in 2015). And considering SOC is continuously decreasing (Fig. 6), the 'realistic' is probably higher than 'RCPs' no matter what scenarios were simulated. Lines 313 and 322: See the above comments. Lines 326-328: Could you please specify the 'biomass changes' and 'SOC variations'; temporal changes (i.e. middle panels in Figs 5 and 6) or spatial changes across the study region. A clearly explanation is necessary for a correct interpretation. Line 329: 'air temperature is the best predictor factor for biomass and SOC' is confusing. Do you want to say 'air temperature is the factor contributing most of the changes or variations in biomass and SOC'. Line 330: Change 'with' to 'have'. Line 332: Discuss section, the authors reported model test results, but did not provide any discussion. It would be good to provide some discussions for all main results. Lines 339-341: In this sentence,

did you still describe the DNDC simulation results? Could you explain why increased temp or precipitation had a positive effect on biomass (Fig 4) while biomass was decreasing from 2015 under the RCPs with temp and precipitation increases. Line 360: What is the meaning of 'the influence of the grassland vegetation dynamics'? Line 366: Change 'with' to 'and'. Line 375: What is the meaning of 'different increase'? Line 383: Change 'improve' to 'increase'. And is this a reason for the simulated increase or just a general knowledge? Line 384: For me, this section is more like Results, instead of Discussion. Line 404: Rewrite this sentence. Do you want to say 'There may have uncertainties for the simulated results due to input uncertainties'? Line 406: What is the meaning of ' the uncertainty of the projected climate will increase the time span'? Line 409: Consider another word to replace 'incorporate'. Line 413: Did you conduct any sensitivity analysis to test the importance of root/shoot ratio? Or did you find any publications to support this point? Lines 417-420: Grammar errors in this sentence, consider rewriting. Line 420: Change 'data' into 'mechanisms'. Line 431: I suggest delete 'slight' because you did not conduct any analysis to investigate if this uncertainty is 'slight' or not. Line 440: This sentence is not a conclusion; I suggest delete this sentence. Table s3: The unit of 'Milk C fraction' et al. should not be %. Table s4: Are these parameters really DNDC default values? Based on my understanding, DNDC does not specify different types of grassland, such as meadow class, alpine steppe et al.

---

## Referee Comment (RC2) · Anonymous Referee #3 · 15 Oct 2017

The comments on the manuscript entitled "Modeling impacts of climate change and grazing effects on plant biomass and soil organic carbon in the Qinghai–Tibetan grasslands" submitted to publish in Biogeosciences. The authors use an ecosystem model to study the changes in SOC and biomass in association with climate change and grazing intensity, which is interesting. The findings of the study are interesting and have important policy and management implications. As the paper is very well written and easy to follow in this revised version compare to the version submitted to the Biogeosciences Discussion. So, my comments are more general. This study seems like it has

the potential to be an important finding, and of broader interest to many constituencies beyond the study areas where the modeling was conducted. So I recommend it can be published in Biogeosciences.

In light of this paper, in a very broad perspective, it is interesting to speculate what happened that results from increasing or decreasing the livestock storage, and it might mean for this region in terms of climate or environmental impacts. Firstly, if livestock increasing, it will raise incomes, lift herdsman out of poverty, but also maybe cause more industrialization in the area? Is the industrialization in this area will induce the grassland utilization method change? I am not sure at all, but there could be long-term environmental impacts of livestock farming development in this region. Secondly, as I know, The study area also the home of a lot of wild animals, whether the livestock numbers decreasing will induce an increasing trend of wild animal population, therefore it will keep the same or higher pressure on the grassland? All these questions are interesting and worth with more studies. But it is likely beyond the scope of the paper to include these in the paper's commentary. Finally, I would like to recommend authors try the manure-DNDC model in the future's work. The manure-DNDC have more detailed simulation process for the livestock system, and you may find more information in Li et al. (2012).

Li, C. et al., 2012. Manure-DNDC: a biogeochemical process model for quantifying greenhouse gas and ammonia emissions from livestock manure systems. Nutrient Cycling in Agroecosystems, 93(2): 163-200.

Some specify minor errors: Line 141 and line 244 with different replicates numbers, please double check. Line 148: Not the "Table s2" in here, please correct. Line 417-420 The sentence is confusing and need to rewrite.
* * *

---

## Author Comment (AC1) · 19 Oct 2017

Replies to Anonymous Referee #3:

Based on the comments and suggestions, we have made careful modifications to the original manuscript. All changes made to the text were with the change tracking in the word. Below you will find our point-by-point responses to the reviewer's comments. Please let us know if you have any questions.

Yours sincerely, Wenjuan Zhang

[Figure]

The comments on the manuscript entitled "Modeling impacts of climate change and grazing effects on plant biomass and soil organic carbon in the Qinghai–Tibetan grasslands" submitted to publish in Biogeosciences. The authors use an ecosystem model to study the changes in SOC and biomass in association with climate change and grazing intensity, which is interesting. The findings of the study are interesting and have important policy and management implications. As the paper is very well written and easy to follow in this revised version compare to the version submitted to the Biogeosciences Discussion. So, my comments are more general. This study seems like it has the potential to be an important finding, and of broader interest to many constituencies beyond the study areas where the modeling was conducted. So I recommend it can be published in Biogeosciences. In light of this paper, in a very broad perspective, it is interesting to speculate what happened that results from increasing or decreasing the livestock storage, and it might mean for this region in terms of climate or environmental impacts. Firstly, if livestock increasing, it will raise incomes, lift herdsman out of poverty, but also maybe cause more industrialization in the area? Is the industrialization in this area will induce the grassland utilization method change? I am not sure at all, but there could be longterm environmental impacts of livestock farming development in this region. Secondly, as I know, The study area also the home of a lot of wild animals, whether the livestock numbers decreasing will induce an increasing trend of wild animal population, therefore it will keep the same or higher pressure on the grassland? All these questions are interesting and worth with more studies. But it is likely beyond the scope of the paper to include these in the paper's commentary. Finally, I would like to recommend authors try the manure-DNDC model in the future's work. The manure-DNDC have more detailed simulation process for the livestock system, and you may find more information in Li et al. (2012). Li, C. et al., 2012.

Manure-DNDC: a biogeochemical process model for quantifying greenhouse gas and ammonia emissions from livestock manure systems. Nutrient Cycling in Agroecosystems, 93(2): 163-200.

[Figure]

Thanks for your kind comments, we agree with that. Link the livestock management with social economic development is a very important research direction, and we recognized that as our future research direction. Indeed, till now, there are still lack the report of considerate both the wild animal and livestock in the simulation, and that is a critical question for the grassland sustainable management. However, as there are lack the detail information of the wild animals distribution and density, therefore, that's a challenging job to simulate both wild animals with livestock together and it is worthwhile to conduct an independent study to explore it.

Some specify minor errors:

Line 141 and line 244 with different replicates numbers, please double check.

Agreed, it's should be 3 replicates. The sentence was corrected.

For each monitoring site, the average value based on 3 replicate sampling points was calculated to determine the aboveground biomass value for the monitoring site.

Line 148: Not the "Table s2" in here, please correct. Line 417- 420 The sentence is confusing and need to rewrite

Agreed, the error was fixed.

Please also note the supplement to this comment:
https://www.biogeosciences-discuss.net/bg-2017-272/bg-2017-272-AC1-supplement.pdf

---

## Author Comment (AC2) · 19 Oct 2017

Replies to Anonymous Referee #2:

Based on the comments and suggestions, we have made careful modifications to the original manuscript. All changes made to the text were with the change tracking in the word (see supplement PDF file). Below you will find our point-by-point responses to the reviewer's comments. Please let us know if you have any questions.

Yours sincerely, Wenjuan Zhang

[Figure]

General comments: This study evaluated responses of grassland ecosystems to both climate change and changes of grazing intensity using a biogeochemical model, DNDC. This topic is important, especially for the region of Qinghai–Tibetan Plateau (QTP), considering that vegetation is sensitive to changes in climate/human management and soil organic carbon (SOC) content is high and potentially easy to loss in this area. This study conducted a regional simulation by coupling the DNDC with a database, and the results provided some useful information regarding future changes of biomass and SOC in QTP. However, I have several concerns about this study.

The first concern is about the model application and the results reported (such as the results in Tables 2, 3 and 4; see the specific points). Detailed introduction regarding how the simulations were implemented and what are the reported values in tables are necessary because these contents are necessary for a correct interpretation. Agreed. A detailed simulation flowchart map was included in the supplement Fig. S1. Moreover, more detailed description of the variables in the Table 2,3was added to the tables footnote section.

The second concern is the statistical analysis in this study. The authors did some statistical analysis to evaluate the simulations. My general feeling is that the statistical analysis could lead to an over-interpretation for the simulated results (if my understanding is correct). In this study, variations of simulations are totally resulted from changes in input parameters (such as climate condition and grazing density), instead of any random factors or instrument errors. Because the authors changed climate condition/grazing density that resulted in the variations, it is not surprising that the biomass and SOC were affected by climate and grazing intensity. So, the descriptions of 'significant effect' are somewhat over-interpretations for me. I suggest the authors rethink about the statistical analysis. The authors may need to clarify that the statistical analysis is not like the statistical analysis for observations with random factors/errors to avoid over-interpretation. Agreed. The sentence of statistical analysis method was rephrased to emphasize the ANOVA analysis in this study was only applied to the simulated results, not the typical field experiments data. Moreover, table 3 and table 4 were combined together to reduce the complexity of the statistical analysis and therefore to avoid over interpretation.

Two-way analysis of variance (ANOVA) was used to test the effects of climate and grazing intensity on both the biomass and SOC according to the simulated results.

Finally, I noticed some inaccurate descriptions, mistakes, and grammar errors. Agreed. All errors were corrected point by point followed the specific point section.

Specific points:

Line 28: Delete 'future'. And I suggest delete 'Thus', because it looks like there is no any causal relationship in these two sentences. Agreed. The sentence was rephrased according to the comment. . . .and RCP8.5 had a more negative effect on the biomass compared with RCP4.5. Future climate change could lead to greater temporal and spatial variations in the grassland biomass and SOC

Line 41: Grammar error in this sentence. Agreed. The sentence rephrased. The Qinghai-Tibetan Plateau (QTP) covers an area of approximately 130 million hectares (ha), 44% of China's total grassland (Li et al., 2013a; Piao et al., 2012).

Line 93: Are these ranges spatial variations or temporal variations of multi-years? Agreed, the range is spatial variations and the sentence was rephrased to clarify the meaning. This region has a typical plateau climate, with a mean annual temperature of 8.6°C (from –6°C to 9°C across the study area) and a mean annual precipitation of 424.7 mm (16.7–776.1 mm across the study area)

Line 121: Delete 'major'. Agreed. The word was deleted. The model has two components.

Lines 123-126: This is not an accurate description. For example, NEE is primarily simulated by tracking vegetation growth and SOC decomposition (instead of nitrification, denitrification or fermentation) in DNDC. Agreed. The sentence was rephrased. The second component includes three submodels for simulating nitrification, denitrification, and fermentation processes, which are used to simulate biogeochemical production, consumption, and emissions of CH4, N2O, NO, and NH3, as well as nitrogen losses due to leaching (Zhang et al., 2015).

Line 148: Here 'Table s2' should be 'Table s4'. Agreed. The number was corrected.

Lines 167-169: Grammar errors in this sentence. Agreed. The grammar errors were fixed. The county boundary data were overlaid on grassland type maps to form the model simulation unit. Then county-based grazing intensity, soil properties, and climate information were assigned to the model simulation units.

Line 179: Are these parameters in the Table s3 DNDC default values, or you determined these parameters based on local information? Parameters in Table s3 were obtained in this study based on DNDC default values and literature reported value.

Line 181: '... for the DNDC grazing model...' should be '... for simulating grass growth'. Agreed. The sentence was rephrased. The detailed parameters for simulating grass growth are shown in Supplementary Table S4

Lines 224-226: Here, could you please specify how did you build the climate data from 2015 to 2044? I notice that there don't have biomass fluctuations between 2015 and 2044, so I guess there is no rainfall fluctuation between 2015 and 2044 (i.e. no dry vs. wet years). Did you use 2014 (or another year) as a 'base', then add temperature and precipitation changes to build climate data from 2015 to 2044. The model's behaviors may be largely regulated by initial or base conditions, so a detailed description on how the climate data were built is necessary. Agreed. More detailed information on how the climate data prepared was added into content.

The third period comprised future climate scenarios (2015–2044), which represented two future climates (RCP4.5ïïjŇRCP8.5) scenarios with changes in temperature and precipitation. The future climate database between 2015 to 2044 was obtained through add the projected future climate change to the daily temperature and precipitation in 2014.

Line 244: '3 replicates' here but '6 replicates' at the Line 141. In addition, it is repeated description of the Lines 140 to 143. Agreed, it's should be 3 replicates. The sentence was corrected and the repeated part was deleted. For each monitoring site, the average value based on 3 replicate sampling points was calculated to determine the aboveground biomass value for the monitoring site.

Line 263 and Fig 2: Total biomass or above ground biomass? In the 'grassland database', you mentioned that 'above ground biomass' is available for model validation. Agreed. It's above ground biomass used to validate the model. The sentence was corrected. There was a significant linear relationship (P < 0.001) between the measurements and the modeled aboveground biomass.

Lines 269-274: This part should be in the 'M&M' section for me. Agreed. The paragraph was removed to M&M section

Lines 289-291: This sentence is general and not informative for summarizing the results of the sensitivity analysis. I suggest delete this sentence. Agreed. The sentence was deleted.

Line 295: Could you please specify the variances in Table 2? Did the variances include both the inter-annual variations during 1985 to 2044 and the variations due to grazing intensity change? Agreed. The variances including both the inter-annual variations during 1985 to 2044 and the variations due to grazing intensity change. As there is no interaction effect between grazing and climate. Therefore the Table 3 and Table 4 were combined and the variances induced by the climate and grazing was reanalyzed with LSD (Least significant difference) values showed in new Table 3.

Lines 300 and 303: The explanations for Tables 3 and 4 are poor. Please clearly explain what are the values in these tables as this influences a correct interpretation. For example, are the values spatial-temporal means across the regions and years (such as 1985-2014 for 'realistic'), with SE representing spatial variations (or other variations)? If so, it may not be fair comparisons between realistic and RCP scenarios because they have different initial conditions (i.e. 'realistic' has a soil condition in 1985 while 'RCPs' have a soil condition in 2015). And considering SOC is continuously decreasing (Fig. 6), the 'realistic' is probably higher than 'RCPs' no matter what scenarios were simulated. Lines 313 and 322: See the above comments. Yes, the values are spatial-temporal means across the regions and years in Table 3 and Table 4, and the SE representing the interannual variations. Indeed, there are the different initial conditions in "realistic" and the "RCPs" scenarios, it not the fair comparisons. According to the reviewer's comments, the table 3 and table 4 were combined to the same table and the simulated values were reanalyzed for more reasonable comparison. The simulated SOC concentrations and total biomass under climate and grazing scenarios were presented in Table 3.

Table 3. The simulated SOC concentrations and total biomass under climate and grazing scenarios. Scenarios Total biomass (g C m–2) SOC (0–20 cm) concentrations (g C kg–1) Climate Realistic (1985–2014) 204.01 66.18 RCP4.5 (2015–2044) 191.17 63.44 RCP8.5 (2015–2044) 183.62 63.37 LSD0.05 3.87 0.09 Grazing Baseline 187.83 64.49 G0 211.42 64.37 G-50 201.41 64.64 G+50 178.11 65.26 LSD0.05 4.47 0.10 LSD0.05: Least significant difference at 0.05 level.

Lines 326-328: Could you please specify the 'biomass changes' and 'SOC variations'; temporal changes (i.e. middle panels in Figs 5 and 6) or spatial changes across the study region. A clearly explanation is necessary for a correct interpretation. Agreed. The sentence was rephrased to clarify the meaning. The biomass changes and SOC variations indicate the annual change value, i.e. the difference value of the biomass and SOC before and end of growing season.

A multiple linear regression analysis was adopted to each simulation unit to analyze the relationship between the annual changed biomass and SOC with corresponding temperature, precipitation and grazing intensity. The regression analysis indicated precipitation, air temperature and combined with grazing intensity, can explain 33.2% of changes in biomass under the realistic climate scenarios with a linear model. Meanwhile, precipitation, air temperature, and grazing intensity can explain 52.3% of SOC variation (Table 4).

Line 329: 'air temperature is the best predictor factor for biomass and SOC' is confusing. Do you want to say 'air temperature is the factor contributing most of the changes or variations in biomass and SOC'. Agreed. The sentence was rephrased. Taking into account the prediction sum of squares (PRESS) value, air temperature is the factor contributing most of the variations in biomass and SOC.

Line 330: Change 'with' to 'have'. Agreed. The word was changed. It's suggested that precipitation and grazing intensity have lower contributes to biomass and SOC change in study region during past thirty years compared to temperature.

Lines 339-341: In this sentence, did you still describe the DNDC simulation results? Could you explain why increased temp or precipitation had a positive effect on biomass (Fig 4) while biomass was decreasing from 2015 under the RCPs with temp and precipitation increases. Agreed. The sentence was the discussion and try to explain why in the sensitivity test in Fig. 4 showed the positive effect on biomass with temp or precipitation increase while biomass was decreasing from the RCPs. Moreover, the sensitivity analysis based on the weather information between (1960-2014), meanwhile, the RCP scenarios climate based on the data in 2014. As there is a significant air temperature increasing trend in the study area during 1957–2008 (Ye et al., 2013), i.e. there is much higher baseline temperature for the model simulation for the RCP scenarios compare with the past 55 years. Therefore, the RCP scenarios simulation showed the decreasing trend with temperature increasing.

Ye, J., Li, W., Li, L. and Zhang, F., 2013. "North drying and south wetting" summer precipitation trend over China and its potential linkage with aerosol loading. Atmospheric

Research, 125(Supplement C): 12-19.

According to this study, the biomass of alpine grassland could increase significantly in the short term as the temperature increases (Fig. 4), as also suggested by Chen et al. (2013) and Gao et al. (2016). However, under long-term constant warming and without considering other meteorological factors, the alpine grassland biomass will probably decrease (Zhu et al., 2016). This may be due to the higher temperature increasing evaporation in the study area, thereby overcoming the benefits of increased precipitation (Xu et al., 2009). The shortage of water will ultimately limit the increase in the grassland biomass with significant warming and drying.

Line 360: What is the meaning of 'the influence of the grassland vegetation dynamics'? Agreed. The sentence was rephrased to clarify the meaning. The grazing intensity is most important for the outcomes of grazing and it is the main external factor that controls the grassland vegetation dynamics.

Line 366: Change 'with' to 'and'. Agreed. The word was changed. ...there was a positive correlation between the grazing intensity and SOC.

Line 375: What is the meaning of 'different increase'? Agreed, the sentence was rephrased to clarify the meaning. Some studies have shown that increasing the plant root/shoot ratio and allocating more carbon to the root system could induce SOC increase.

Line 383: Change 'improve' to 'increase'. And is this a reason for the simulated increase or just a general knowledge? Agreed. The word was changed. The is one of the possible reasons for the simulated increase of SOC. Moreover, increases in the effects of hoof activity can accelerate the decomposition of litter and decaying roots, and improve the contact with the soil, thereby accelerating the transfer of carbon to the soil to increase the SOC concentration (Luo et al., 2010; Naeth et al., 1991).

Line 384: For me, this section is more like Results, instead of Discussion. Agreed, this section was moved to the results part.

Line 404: Rewrite this sentence. Do you want to say 'There may have uncertainties for the simulated results due to input uncertainties'? Agreed. The sentence rephrased. However, the uncertainty of the data sources could be incorporated into the model outputs.

Line 406: What is the meaning of ' the uncertainty of the projected climate will increase the time span'? Agreed. The sentence was rephrased to clarify the mean. The CMIP5 RCP scenarios were used to provide the possible changes in climate in this study, but as a long-term climate projection, the uncertainty of the projected climate will increase with time span increase.

Line 409: Consider another word to replace 'incorporate'. Agreed. The word was changed. . . .this assumption may cause uncertainty for long-term study.

Line 413: Did you conduct any sensitivity analysis to test the importance of root/shoot ratio? Or did you find any publications to support this point? Agreed. As we didn't implement the root: shoot ratio sensitivity test and this sentence was deleted.

Lines 417-420: Grammar errors in this sentence, consider rewriting. Agreed. In the present study, we assumed that the grassland type was the same in the scenarios. As the grassland community structure could be altered under both grazing and climate change (Koerner and Collins, 2014). Therefore, the assumption of grassland community structure keeps stable in the simulation could induce the uncertainty.

Line 420: Change 'data' into 'mechanisms'. Agreed. The word was changed. Due to a lack of mechanisms regarding the response of grassland soil to animal trampling in the DNDC model. . .

Line 431: I suggest delete 'slight' because you did not conduct any analysis to investigate if this uncertainty is 'slight' or not. Agreed. The word was deleted. These assumptions could have induced slight uncertainties in the simulation results.

Line 440: This sentence is not a conclusion; I suggest deleting this sentence. Table s3: The unit of 'Milk C fraction' et al. should not be %. Table s4: Are these parameters really DNDC default values? Based on my understanding, DNDC does not specify different types of grassland, such as meadow class, alpine steppe et al. Agreed. The unit was corrected and the sentence was deleted. The fraction parameters were unit less and with a range between 0-1. The parameters for the different types of grassland in Tables s4 was derived from the observed data and literature information. In the model simulation, the corresponding grassland parameters based on the grassland map of the study area will be applied.

Please also note the supplement to this comment:
https://www.biogeosciences-discuss.net/bg-2017-272/bg-2017-272-AC2-supplement.pdf

**Supplement:**

[revised manuscript text omitted]

Dong Quan min and quan, Z. X.: Effects of Grazing Intensity and Time on Dry Matter Intake and its Apparent Digestibility of Yaks, China Herbivores, 27, 12-14, 2007.

ji, W. Q., min, Z. X., qing, Z. Y., and xi, S. Z.: Community structure and biomass dynamic of the Kobresia pygmaea steep meadow, Acta Phytoecologica Sinica, 19,, 225-235, 1995.

li, Z. L., hua, S. M., feng, Y. T., and hai, Y. F.: Feeding preference of Gynaephora menyuanensis and its relationships with plant carbon and nitrogen contents in an alpine meadow on the Tibetan plateau, ACTA ECOLOGICA SINICA, 2016. 2319-2326, 2016.

lin, W. J., ming, Z. Z., hong, W. Z., xiong, C. B., qun, Y. C., xiang, H. X., xi, S. Z., Dacizhuoga, and zhou, Z. X.: Soil C/N distribution characteristics of alpine steppe ecosystem in Qinhai-Tibetan Plateau, ACTA ECOLOGICA SINICA, 2014. 6678-6691, 2014.

QPBS: 'Qinghai Statistical Yearbook 2015.'(China Statistics Press:Beijing), 2015. 2015.

Xue Bai, Zhao Xin quan, and sheng, Z. Y.: Feed Intake Dynamic of Grazing Livestock in Nature Grassland in Qinghai-Tibetan Plateau, Ecology of Domestic Animal, 25, 21-25, 2004.

Zhang, B.: On the Livestock Development of Qinghai Province during the Time of Republic of China(1912—1949), Ancient and Modern Agriculture, 2011. 91-100, 2011.

[Figure]

Fig. S1.   The flowchart of model data preparation and simulation. The vector and raster inside the brackets indicate the input data format, and the intersection and zonal statistic inside the rhombus indicate the ArcGIS algorithm applied to process the data.

[Figure]

Fig. S2. Spatial distributions of precipitation and changes in temperature under the RCP4.5 (a, c) and RCP 8.5 (b, d) climate change scenarios up to 2044.

---

## Author Response (AR2)

Dear Kuzyakov:

Thanks for your valuable comments, we have made careful modifications to the manuscript following your suggesting. Moreover, we had cited more relative works implemented in this area and organized all references followed the journal's requirement. All changes made to the text were with the change tracking in the word. Please let us know if you have any questions.

Yours sincerely,

Wenjuan Zhang and Fujiang Hou

Dear Authors,

thank you for the response to the reviewers and incorporation of their suggestions.
I read the Abstract and looked on the Tables and Figures,
and to regret - the Conclusions and the main outcomes of the paper are not clear. So, despite a lot of work was done, the outcome / conclusions are not clear, or mainly described by general words, that can be written also without to do such a lot of work.
So, I request you to put in the abstract clear and quantitative conclusions to: 1) climate effects, 2) grazing effects and this on 3) plant biomass and 4) SOC. Your conclusions should be quantitative and all general words should be removed. Also the relevance of the conclusions should be presented.
Please add / improve this and your paper will get much more attraction of the readers.

Sincerely yours,
Yakov Kuzyakov

Agreed. The abstract and the conclusion paragraph was reorganized.

[revised manuscript text omitted]